# Copy number signature analysis tool and its application in prostate cancer reveals distinct mutational processes and clinical outcomes

Shixiang Wang[1,2,3], Huimin Li[1,2,3], Minfang Song[1,2,3], Ziyu Tao[1,2,3], Tao Wu[1,2,3], Zaoke He[1,2,3], Xiangyu Zhao[1,2,3], Kai Wu[4], Xue-Song Liu[1] *

1 School of Life Science and Technology, ShanghaiTech University, Shanghai, China, 2 Shanghai Institute of Biochemistry and Cell Biology, Chinese Academy of Sciences, Shanghai, China, 3 University of Chinese Academy of Sciences, Beijing, China, 4 Department of Thoracic Surgery, The First Affiliated Hospital of Zhengzhou University, Zhengzhou, China

* liuxs@shanghaitech.edu.cn

**Data Availability Statement:** The authors confirm that all data underlying the findings are fully available without restriction. All relevant data are within the paper and its Supporting Information

## Abstract

Genome alteration signatures reflect recurring patterns caused by distinct endogenous or exogenous mutational events during the evolution of cancer. Signatures of single base substitution (SBS) have been extensively studied in different types of cancer. Copy number alterations are important drivers for the progression of multiple cancer. However, practical tools for studying the signatures of copy number alterations are still lacking. Here, a user-friendly open source bioinformatics tool "sigminer" has been constructed for copy number signature extraction, analysis and visualization. This tool has been applied in prostate cancer (PC), which is particularly driven by complex genome alterations. Five copy number signatures are identified from human PC genome with this tool. The underlying mutational processes for each copy number signature have been illustrated. Sample clustering based on copy number signature exposure reveals considerable heterogeneity of PC, and copy number signatures show improved PC clinical outcome association when compared with SBS signatures. This copy number signature analysis in PC provides distinct insight into the etiology of PC, and potential biomarkers for PC stratification and prognosis.

## Author summary

Genomic DNA alteration signatures are recurring genomic patterns that are the imprints of mutagenic processes accumulated over the lifetime of cancer cell. Copy number alteration is a key driver for the progression of multiple cancer, including prostate cancer, which is particularly driven by complex genome alterations. However, practical tools for studying the signatures of copy number alterations are still lacking. Here a novel bioinformatics tool for copy number signature analysis has been constructed. With this newly developed bioinformatics tool "sigminer", we performed the first copy number signature analysis in prostate cancer, and an unprecedentedly clear map connecting genome alteration driving factors and prostate cancer clinical outcomes have been illustrated. These

files. All code required to reproduce the analysis outlined in this manuscript are freely available at https://github.com/XSLiuLab/PC_CNA_signature. Analyses can be read online at https://xsliulab. github.io/PC_CNA_signature.

**Funding:** This work was supported by The National Natural Science Foundation of China (http://www. nsfc.gov.cn): 31771373 [X.-S.L.]. The funders had no role in study design, data collection and analysis, decision to publish, or preparation of the manuscript.

**Competing interests:** The authors have declared that no competing interests exist.

analyses provide novel insight into the mutational processes and clinical outcomes of prostate cancer.

## Introduction

Cancer are primarily caused by somatic alterations in the genomic DNA. Based on the size and feature of genome alterations, these cancer associated DNA alterations can be classified into the following four types: single base substitution (SBS), small insertion and deletion (INDEL), structural alteration including translocation/inversion, and copy number alteration (CNA). Somatic copy number alterations are extremely common in cancer, and have been reported as important drivers for the progression of multiple types of cancer [1,2].

Genomic DNA alteration signatures are recurring genomic patterns that are the imprints of mutagenic processes accumulated over the lifetime of cancer cell [3,4]. Genome alteration signature analysis can not only provide the mutational process information, but also biomarker for cancer precision medicine [5,6]. SBS signature analysis has been extensively studied, and represents a prototype for other types of signature study [3]. Despite the importance of copy number alteration in cancer progression, practical tools for copy number signature study are still lacking.

Prostate cancer (PC) is a common cancer type among men [7]. PC is particularly driven by copy number alterations, indolent and low-Gleason tumors have few alterations, whereas more aggressive primary and metastatic tumors have extensive copy number alterations [8,9]. In contrast, somatic point mutations are less common in PC than in most other solid tumors [10]. The most frequently mutated genes in primary PC are *SPOP*, *TP53*, *FOXA1*, and *PTEN* [11]. TMPRSS2-ETS fusion genes are frequently observed in prostate cancer patients [12]. Several studies reported that percent of copy number altered genome, termed "CNA burden" is associated with the recurrence and death of prostate tumors [9] and other tumor types [13]. Aneuploidy, defined as chromosome gains and losses, has been reported to drive lethal progression in prostate cancer [14]. All these results suggest a key role of CNA in PC progression and clinical outcome association. However, both CNA burden and aneuploidy only capture particular aspect of CNA, comprehensive understanding of CNA in PC remains obscure, and the underlying mechanisms and the specific mutational processes involved also remain unknown.

To address these challenges, we developed a novel method to investigate the signatures of copy number alteration, and built an open source R/CRAN package for the scientific community. With this tool, we performed the first copy number signature analysis in PC. The driving forces and mechanisms underlying each distinct copy number signature have been proposed, and the clinical outcome for each signature has been further investigated. This copy number signature analysis provides a new insight into the mutational processes of prostate tumors, and also novel biomarkers for PC stratification and prognosis.

## Results

### Copy number signature analysis framework

SBS mutations are primarily the consequences of single-strand DNA lesions and single-strand break repair mistakes. Copy number alterations reflect double-strand DNA lesions and double-strand break repair problems. SBS signatures have been well studied in many cancers, and summarized in the COSMIC (Catalogue of Somatic Mutations in Cancer) database [4].

However, copy number signatures are less studied. Macintyre et al. performed copy number signature analysis in high-grade serous ovarian cancer [15], and represented the currently only available study related to copy number signature. Macintyre et al. employed a mixture modeling based method for copy number component extraction, the biological meaning of the copy number component is unclear and not consistent among different cancer types or datasets, this limits the application of their method in cancer genome study [15]. A unified and extensible copy number component classification method and practical bioinformatical toolkit are required.

Here a unified copy number signature extraction method has been constructed (Fig 1). It incorporated the following 8 copy number features: the breakpoint count per 10 Mb (named "BP10MB"); the breakpoint count per chromosome arm (named "BPArm"); the absolute copy number of the segments (named "CN"); the difference in copy number values between adjacent segments (named copy number change point, or "CNCP"); the lengths of oscillating copy number segment chains (named "OsCN"); the log10 based copy number segment size (named "SS"); the minimal number of chromosomes with 50% copy number alterations (named "NC50"); the distribution of copy number alterations in each chromosome (burden of chromosome, named "BoChr") (Fig 1). These features were selected as hallmarks of previously reported genomic aberrations like chromothripsis, tandem duplication or to denote the genome distribution pattern of copy number alteration events [15–17].

We classified the distributions of the above mentioned 8 copy number features into 80 components (Fig 1 and S1 Table). Each copy number component has a clear biological meaning, for example: "CN[2]" indicate the absolute copy number of the DNA segment is 2. For each tumor, the value for each component of copy number feature was counted based on the segmented absolute copy number profile of that tumor. The absolute copy number profiles can be extracted from whole exome sequencing (WES), whole genome sequencing (WGS), or SNP array data [18,19]. Then a tumor by copy number component value matrix was generated by combining component values in all tumors. This matrix was subjected to non-negative matrix factorization (NMF), a method previously used for deriving SBS signatures [3].

Compared to Macintyre et al. method [15], our method is much more computationally efficient, it took about 1.5 minutes with our method rather than about 1.5 hours using Macintyre et al. method to generate the matrix for copy number signature extraction in this study. Besides, each of the predefined 80 copy number components has clear and fixed biological meaning, and this facilitates not only the intelligibility of signature result, but also the scalability of this copy number signature analysis method with known SBS signature analysis. The biological meaning of the CNA component in Macintyre et al. method is not consistent for different datasets. This is problematic especially if the investigator want to compare the signature generated with different dataset using cosine similarity analysis. To illustrate this difference, we compared our method with Macintyre et al. method in ovarian cancer with different sample size. The CNA component is consistent for three datasets with our method but not Macintyre et al. method, consequently the signatures in different dataset cannot be compared when using Macintyre et al method (S1 Fig). Due to this problem Macintyre et al. method cannot be applied in single sample CNA signature analysis. The major differences between our method and Macintyre et al. method are listed in S2 Fig.

A R/CRAN package (Sigminer, https://cran.r-project.org/package=sigminer) has been developed based on this novel copy number analysis method for bioinformatics community and cancer researchers to explore and analyze copy number signatures (Fig 1). To our knowledge, sigminer is the first practical bioinformatics tool for extracting the signatures of copy number alterations, supporting both our new method and the method from Macintyre et al. study [15].

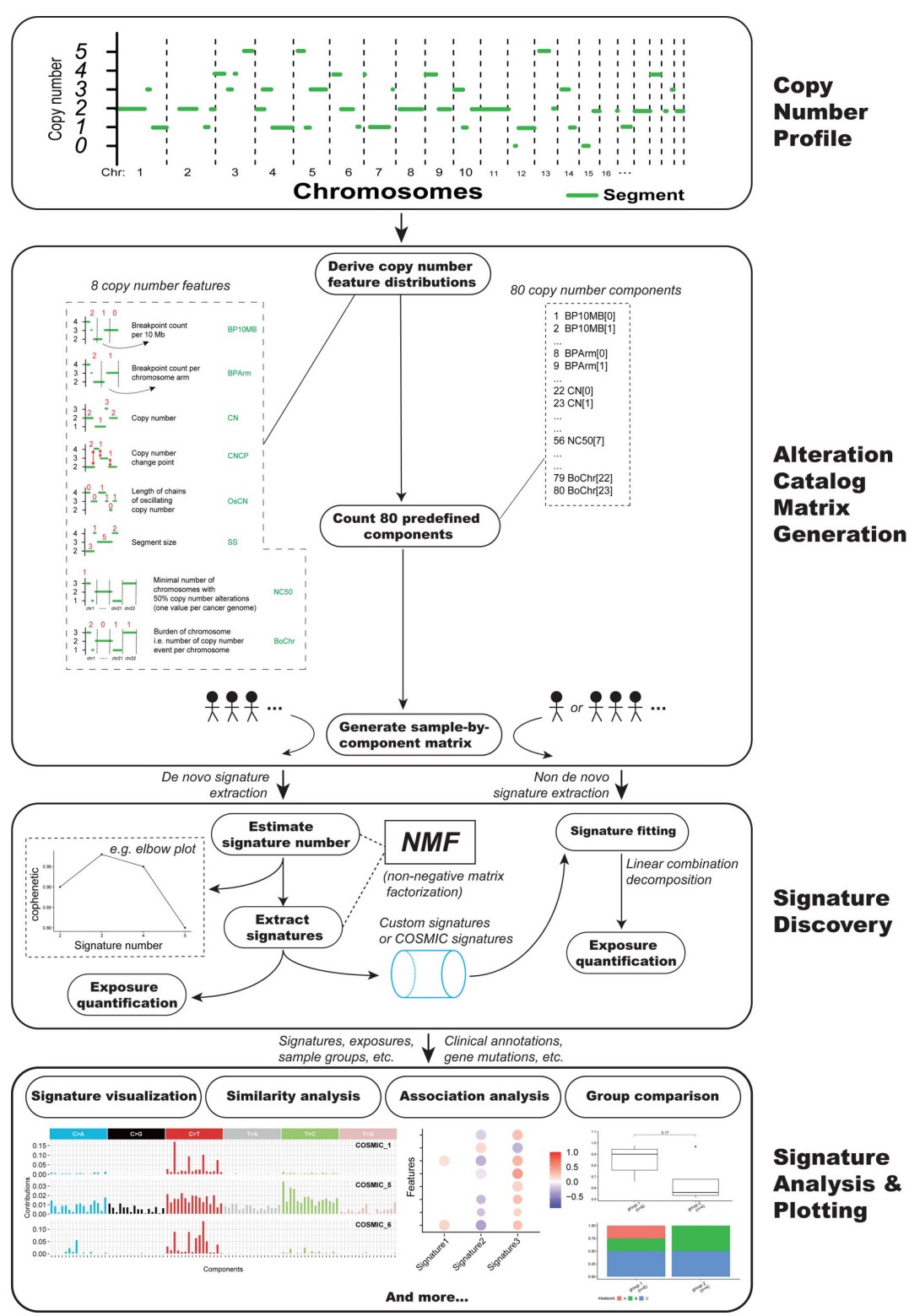

**Fig 1. Copy number signature analysis framework.**

## Genome alteration landscape of PC

We assembled and uniformly analyzed WES data from 1,003 pairs of prostate cancers and matched germline control (649 primary and 354 metastatic tumors) that passed quality control parameters from six independent studies (S3 Fig) [10,20–24]. Patient characteristics, including age at diagnosis, Gleason score, and metastatic site, are shown in S2 Table.

Small scale genome alterations include single base substitution (SBS) and small insertion and deletion (INDEL). Mutation significance analysis with these small scale genome alterations revealed 47 cancer driver genes (MutSig q < 0.05) (S4 Fig). Similar to previous study [11], the top frequently mutated genes in this PC genome dataset are *TP53* (19%), *SPOP* (9%), *FOXA1* (8%), *PTEN* (4%) (S4 Fig). About half of PC samples (56.44%, 552 of 978) have small scale cancer driving genome alterations. These small scale genome alterations were summarized based on different variant classifications (S5A–S5C Fig). The median number of small scale variants in PC is 28, and most of them are SBS, suggesting a relatively low SBS number in PC compared with other cancer types (S5B Fig).

Distribution pattern of CNA segment length and chromosome distribution of CNA counts in available PC samples are summarized and shown (S5D Fig). Majority of copy number alterations are focal, and 14.9% are chromosome arm level or whole chromosome level alterations (S5D Fig). Median values of CNA segment count, CNA amplification count, CNA deletion count and CNA burden are shown in S3 Table. The distribution of 8 CNA features selected in this study are shown (S5E Fig).

## Genome alteration signatures identified in PC

With sigminer developed in this study, we identified five copy number signatures and three SBS signatures from 1003 tumor-normal pairs of PC WES data. The number of signature was determined after comprehensive consideration of result stability shown by cophenetic plot and biological interpretability (S6 Fig).

Three SBS mutational signatures named SBS-sig 1, SBS-sig 2, SBS-sig 3 are identified (S7A Fig). Etiology for the SBS mutational signatures has been well explored and stored in COSMIC database. Cosine similarity analysis has been performed between the three SBS signatures identified in this study and COSMIC signatures (S8 Fig). The SBS signatures identified in this study are similar to those identified by COSMIC based on TCGA PC datasets.

Five copy number alteration signatures are identified, namely CN-sig 1 to CN-sig 5 (Fig 2A). These copy number signatures are ranked based on the median length of CNA segment. CN-sig 1 has the smallest median size of CNA segment, and CB-sig 5 has the largest median size of CNA segment. Copy number components of the same feature are row normalized, and this facilitates copy number component value comparisons within the signature. Macintyre et al. performed column normalization for each copy number component, and the component values of same CNA feature cannot be compared within each signature [15]. Representative absolute copy number profile for each signature enriched PC patient is shown (Fig 2B).

## Signature stability and single sample copy number signature analysis

In cancer precision medicine, it is important to confidently associate known signatures and their activities to patients. One of the key issues in signature analysis is the stability of signature extracted. Bootstrap analysis as previous reported [25] was performed to evaluate the stability of signatures extracted through *do novo* NMF (S7B Fig). Comparison analysis of signature exposure instability between all signatures has been applied. Every signature instability measured as the root mean squared error (RMSE) between its exposures in 1000 bootstrap mutation catalogs and its exposures in the original mutation catalog for each tumor are shown (Fig

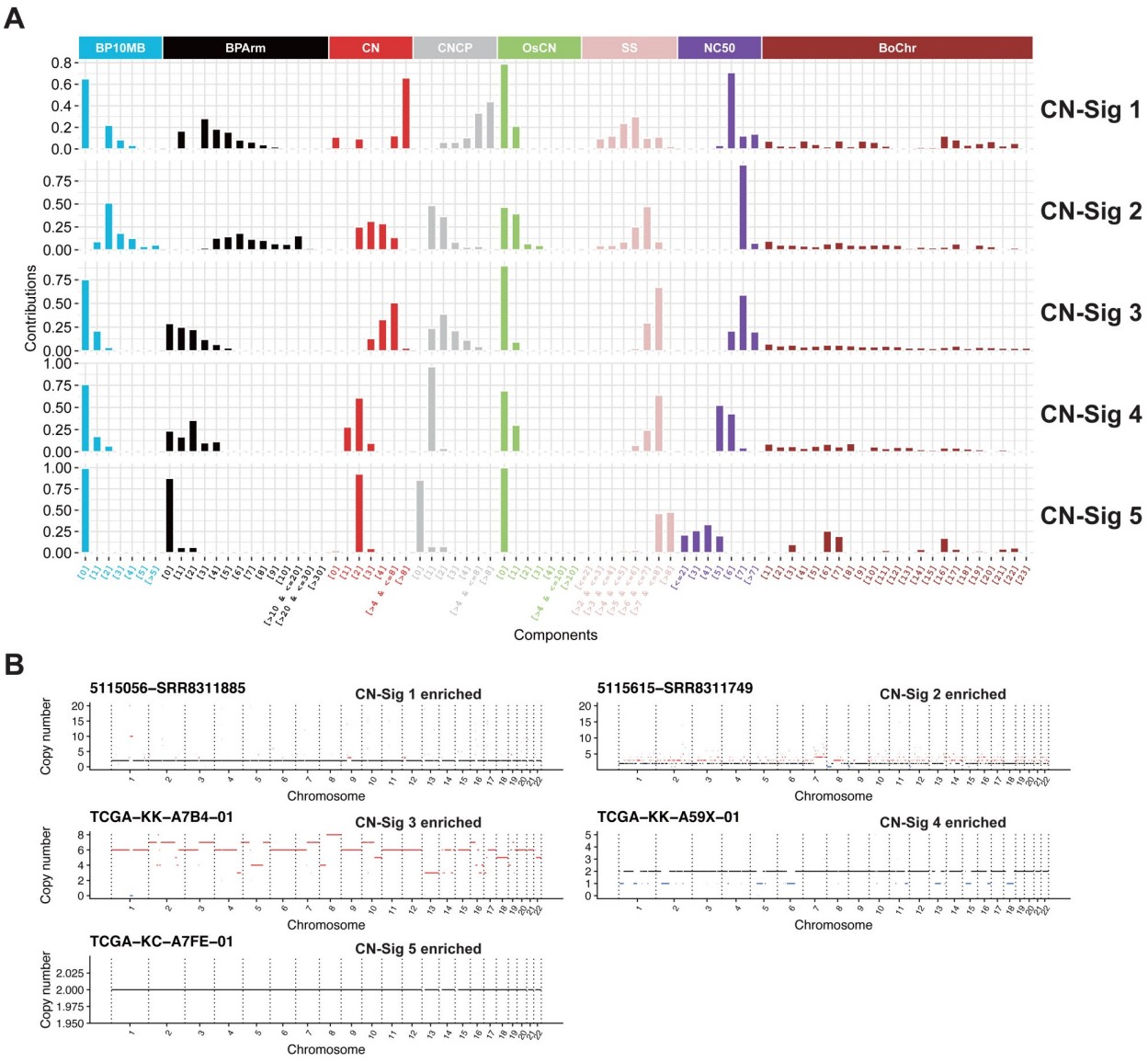

**Fig 2. Copy number signatures identified in PC.** (A) Five copy number signatures are identified from PC WES data. For copy number signatures, the signature profile was row normalized within each feature. Copy number signature has 8 different features with 80 components in together. (B) Representative absolute copy number profile for each copy number signature. Selected samples with enriched copy number signature CN-Sig 1, CN-Sig 2, CN-Sig 3, CN-Sig 4 and CN-Sig 5 are shown. The segments with copy number gain and loss are labeled by red and blue color, respectively.

3A). For the combined PC WES dataset, at least 57 tumors are required for extraction of all five copy number signatures (Fig 3B).

In addition to *de novo* copy number signature extraction with NMF, method for single sample copy number signature analysis has also been developed based on signature fitting and optimization with simulated annealing [25]. As an example, the five PC samples in Fig 2B are reanalyzed with single sample signature fitting method. Each tumor is clearly dominated by the same type of copy number signature as the signature derived from *de novo* NMF (Fig 3C). Bootstrap procedure was also adopted to evaluate the stability of signatures extracted through single sample copy number signature fitting in two different samples, and consistent results

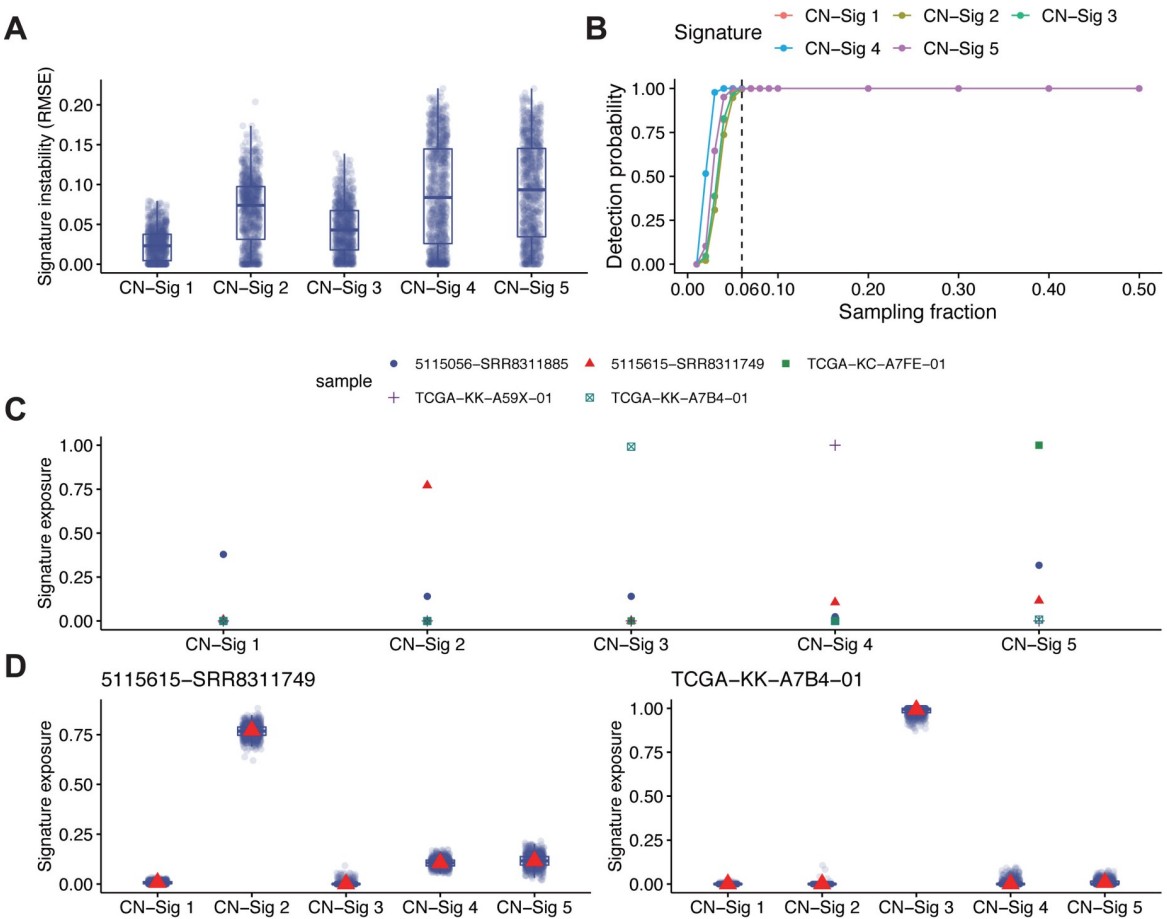

**Fig 3. Signature instability analysis and single sample copy number signature analysis.** (A) Comparison of signature exposure instability measured as RMSE (root mean squared error) between exposures in 1000 bootstrap mutation catalogs and exposures in the original mutation catalog for each tumor. (B) Signature detection probability versus different sampling fraction. The dotted line indicates we could steadily detect all five copy number signatures with at least 6% (~57) tumors. A signature is detected if more than 10 tumors have at least 1% relative exposure under p-value cutoff 0.05. The sampling process was repeated 1000 times for each sampling fraction to estimate the detection probability. (C) Relative signature exposures measured through single sample signature fitting for the five tumors in Fig 2B. (D) Single sample signature instability analysis for two of the five tumors in Fig 2B based on 1000 bootstrap signature exposures (indicated as blue dots and boxplot) and the original signature exposure (indicated as red triangle). The boxplot is bounded by the first and third quartile with a horizontal line at the median.

are obtained (Fig 3D). All these analyses suggest that the copy number signature extraction method developed in this study can provide reliable and consistent results.

## Mutational processes underlying copy number signatures

Five copy number signatures and three SBS signatures have been identified in 1003 tumor-normal paired PC WES dataset. These signatures reflect distinct genomic DNA alteration patterns driving by distinct molecular events. To investigate the interconnection among genome alteration signatures, we computed the associations between the exposures of signatures and various types of genomic alteration features including tumor purity, MATH (A simple quantitative measure of intra-tumor heterogeneity [26]), tumor ploidy, CNA burden, INDEL number, tandem duplication phenotype score (TDP score, see methods) [17], chromothripsis state score (see methods) [16] (Fig 4A), genetic alteration in known PC driving genes (Fig 4B), and PC associated signaling pathways (Fig 4C and S4 Table).

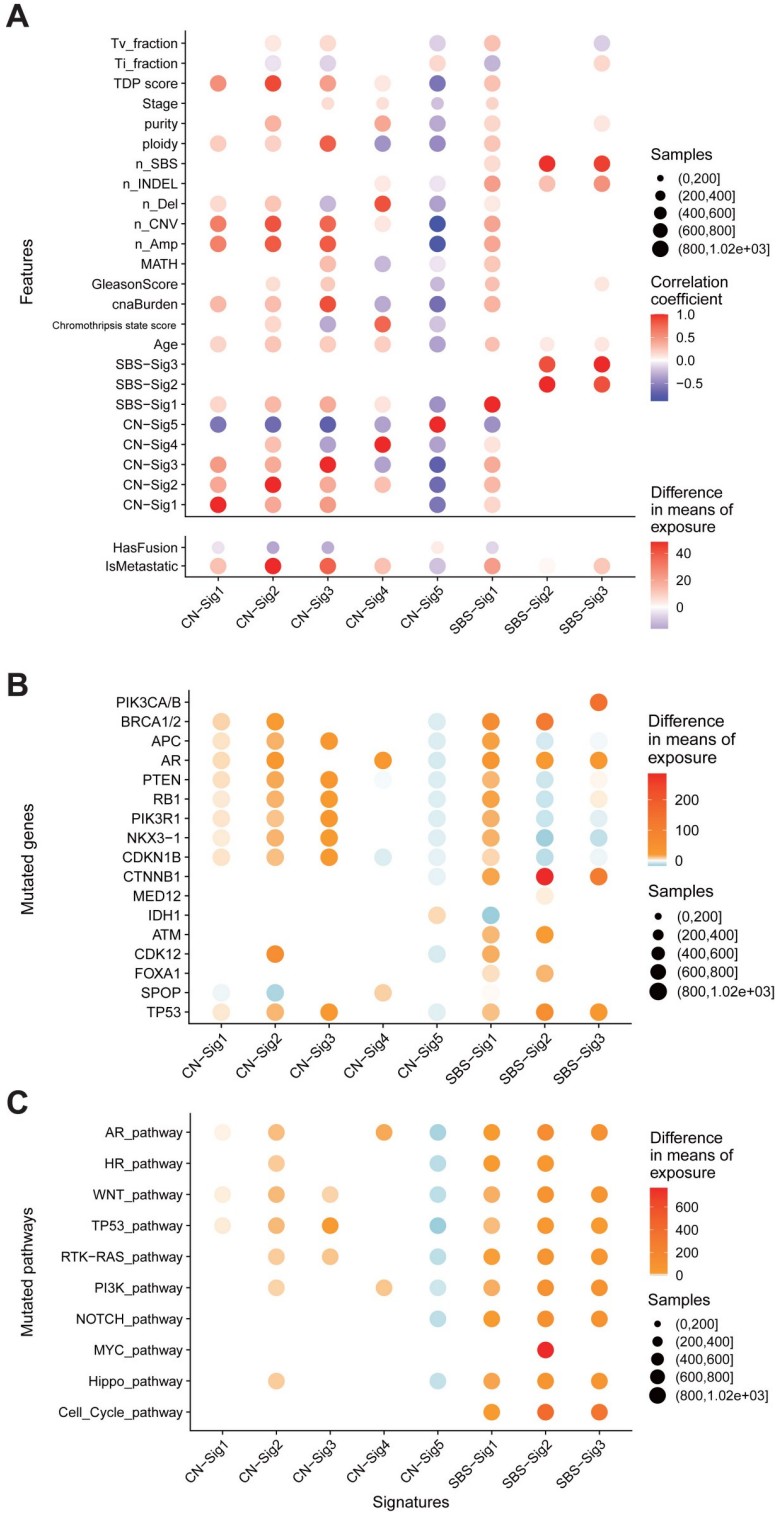

**Fig 4. Genome alteration signatures and mutational processes in PC.** (A) Associations between the exposures of genome alteration signatures and features including clinical parameters, somatic DNA alteration quantifications. (B,C) Associations between the exposures of genome alteration signatures and mutations in selected genes (B) or pathways (C). Only differences with false discovery rate P < 0.05 (Mann–Whitney U-test) are shown. Red, positive correlation; blue, negative correlation. The depths of colors in filled circles indicate the extent of difference. Sizes of circle in all plots indicate the number of cases included in each analysis.

**Table 1. Genome alteration signatures in PC.** Associations for each genome alteration signature and proposed mechanisms are shown.

| Signature | Proposed mechanism or etiology | Evidences | Notable associations |
|---|---|---|---|
| *CN-Sig 1* | Focal amplification | Many short, focally localized DNA segments with very high absolute copy number | - |
| *CN-Sig 2* | Tandem duplication | Many evenly distributed medium-length DNA segment amplification; high number of breakpoints | Positive association: *CDK12* mutation; Homologous recombination pathway mutation; TDP score; metastasis. Negative association: SPOP mutation; *TMPRSS2–ETS* fusion. |
| *CN-Sig 3* | Whole-genome duplication | High ploidy, High absolute copy number; Few OsCN; global CNA distribution | Positive association: *TP53* mutation; CNA burden; ploidy; Negative association: *TMPRSS2–ETS* fusion; CN-sig 4; CN-sig 5. |
| *CN-Sig 4* | Chromothripsis | Copy number change point is one; considerable OsCN | Positive association: *AR* mutation; *SPOP* mutation; copy number deletion number; chromothripsis state score. Negative association: CN-Sig3; MATH. |
| *CN-Sig 5* | Copy number neutral like events including oncogenic *TMPRSS2–ETS* fusion | Few CNA; Focal distribution of CNA | Positive association: *TMPRSS2–ETS* fusion; *IDH1* mutation; Negative association: mutations (except *IDH1* mutation) and other genome alterations. |
| *SBS-Sig 1* | Homologous recombination defect or unknown | Highly similar to COSMIC SBS signature 3/5 | Positive association: copy number signatures. |
| *SBS-Sig 2* | Mismatch repair defect | Highly similar to COSMIC SBS signature15/6 | - |
| *SBS-Sig 3* | Aging | Highly similar to COSMIC SBS signature 1 | - |

CN-sig 5 shows negative association with nearly all cancer genome alteration features excluding *TMPRSS2-ETS* oncogenic fusions. And CN-sig 5 is the only genome alteration signature that shows positive correlation with the presence of *TMPRSS2-ETS* oncogenic fusions. *TMPRSS2-ETS* fusion happens in early stage of PC [27]. All these observations suggest that CN-sig 5 reflects a stable genome or early PC evolving state.

Multidimensional clustering with genome alteration signatures and PC clinical parameters results in several clusters (S9 Fig). CNA burden feature is clustered with CN-sig 1, 2, 3, 5. And in this cluster, CN-sig 5 show negative correlation with other features or signatures. CN-sig 4 is associated with number of deletions, and forms a distinct cluster when compared to other copy number signatures. SBS signatures form separate clusters from copy number signatures. SBS-sig 2 and SBS-sig 3 form cluster with number of INDEL and total mutation. SBS-sig 1 form a different cluster. These analyses revealed underlying connections between different genome alteration signatures.

Based on the distribution of the value of each copy number component (Fig 2), copy number profiles of representative tumors (Fig 2B), and signature correlation analyses (Figs 4 and S9), the etiologies and mechanisms for copy number signatures has been proposed as following (Table 1):

CN-sig 1 is represented by many short (0.01–0.1Mb), focally localized DNA segments with very high absolute copy number (>8). This signature is caused by focal amplification of DNA segments.

CN-sig 2 is represented by evenly or globally distributed, large amount of medium-length (0.1–10 Mb) DNA segment amplifications. This signature is associated with *CDK12*, homologous recombination (HR) pathway gene mutations and high TDP score, and shows highest association with PC metastasis. This signature can be a result of defective HR DNA repair and consequently tandem duplication phenotypes.

CN-sig 3 is represented by high tumor ploidy, and is associated with high CNA burden and *TP53* mutation. This signature reflects the occurrence of whole genome doubling events.

CN-sig 4 is featured by DNA segment one copy deletion, and considerable oscillating copy number. This signature is associated with *AR*, *SPOP* mutation and high chromothripsis state score, suggesting a state of chromothripsis [28].

CN-sig 5 is represented by large copy number segment size with few background copy number alterations, and is the only signature positively associated with PC oncogenic fusions. This signature is also the only copy number signature enriched specifically in non-metastatic PC, and is associated with good survival. This signature is a result of relatively stable PC genome, reflecting copy number neutral mutation processes (e.g. *TMPRSS2-ETS* oncogenic fusions).

## Copy number signature and PC patients' stratification

PC samples are clustered into five groups based on the consensus matrix from multiple NMF runs, and each group is specified by one enriched copy number signature (Fig 5A). Clinical and genomic parameters including patients' age, SBS count (n_SBS), number of insertion/ deletions (n_INDEL), transition (Ti) fraction, transversion (Tv) fraction, CNA number, copy number amplification number (n_Amp), copy number deletion number (n_Del), CNA burden, tumor purity, and tumor ploidy are compared in each copy number signature enriched PC patient group (Fig 5B). Significant differences are observed in all above mentioned clinical parameters, except total mutation and number of INDEL among different copy number signature enriched PC groups (Fig 5B). Similar analysis was performed using SBS signature to cluster PC patients. Most clinical and genomic parameters do not show significant difference among three SBS signature enriched PC groups (S10 Fig). This analysis suggests that copy number signatures could have more PC stratification power when compared with SBS signatures.

Metastasis is the leading cause of PC associated death, the associations between each genome alteration signatures and PC metastasis are shown as Sankey diagram (S11A Fig). CN-Sig 2 is nearly exclusively found to be associated with PC metastasis, and CN-Sig 5 is nearly exclusively found not to be associated with PC metastasis. Compared to copy number signatures, SBS signatures do not show strong associations with PC metastasis. This metastasis association analysis suggests that copy number signature can be PC metastasis specific biomarker, and this analysis further demonstrates that copy number signatures provide more PC stratification information than SBS signatures.

The associations between copy number signatures and metastasis, Gleason score and clinical stage have been statistically analyzed (S11B Fig). Similar analysis has been performed with SBS signatures (S11C Fig). Significant differences in metastasis, Gleason score and clinical stage status are observed among different copy number signature enriched PC patients. However, among SBS signature enriched PC patients, no significant differences are observed. These analyses again demonstrate that copy number signatures carry more PC stratification information than SBS signatures.

## Copy number signatures predict PC patients' survival

Univariate Cox regression analyses were performed to evaluate the associations between each genome alteration signature and PC patients' survival time (Fig 6). In overall survival (OS) analysis, only copy number signatures show significant correlations, none of SBS signatures show significant correlations (Fig 6A). CN-sig 2 is significantly associated with poor OS, and CN-sig 5 is significantly associated with improved OS. In regards to progression-free survival (PFS), CN-sig 3 is significantly associated with poor PFS, and CN-sig 5 is significantly associated with improved PFS (Fig 6B).

**A**

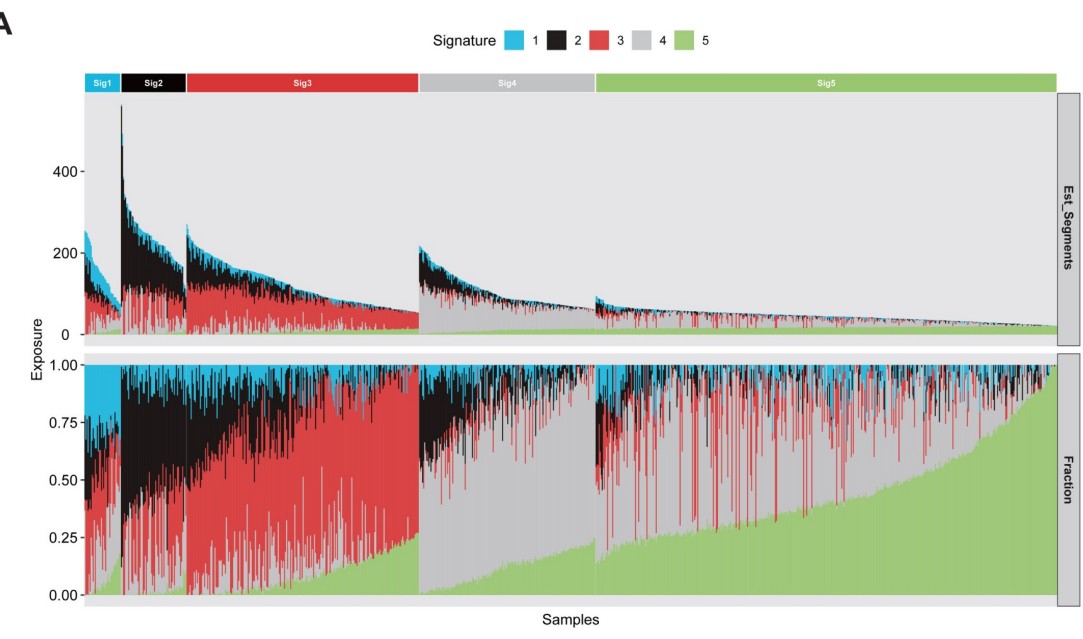

**B**

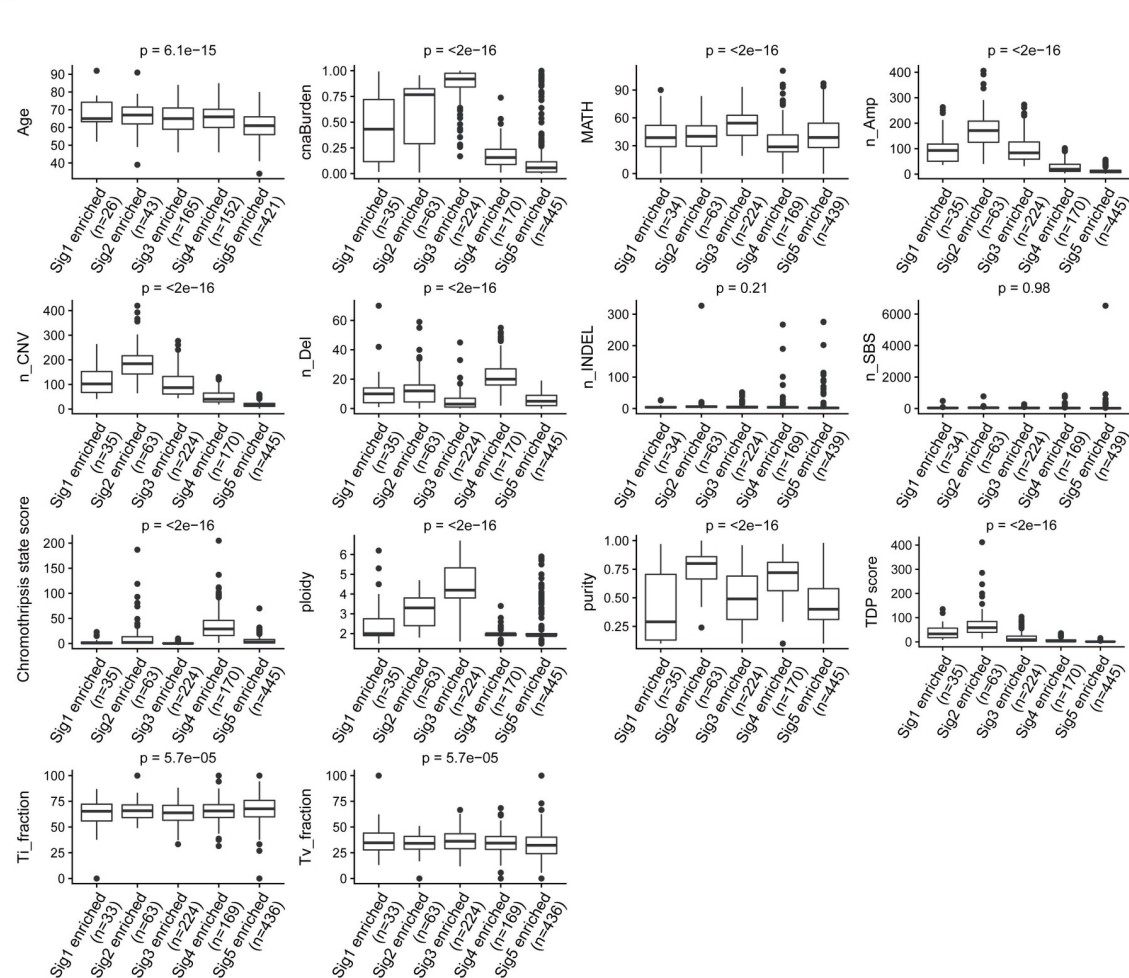

**Fig 5. Sample clustering and heterogeneity analysis of PC based on the exposures of copy number signatures.** (A) For each PC patient, the relative contribution (bottom panel) and estimated copy number segment counts (top panel) of each signature are shown as a staked barplot. PC samples are clustered into five groups based on the consensus matrix from multiple NMF runs, and each group is specified by one enriched copy number signature. (B) Quantitative comparison for somatic and clinical parameters among each copy number signature enriched PC group by boxplot. The boxplot is bounded by the first and third quartile with a horizontal line at the median. ANOVA p values are shown. Abbr.: n_, number of, e.g. n_SNV, number of SNV; INDELs, small insertions and deletions; Ti_fraction, transition fraction; Tv_fraction, transversion fraction; Amp, copy number segments with amplification; Del, copy number segments with deletion; TDP score, tandem duplication phenotype score; cnaBurden, copy number alteration burden; MATH, MATH score is a quantitative measure of intra-tumor heterogeneity.

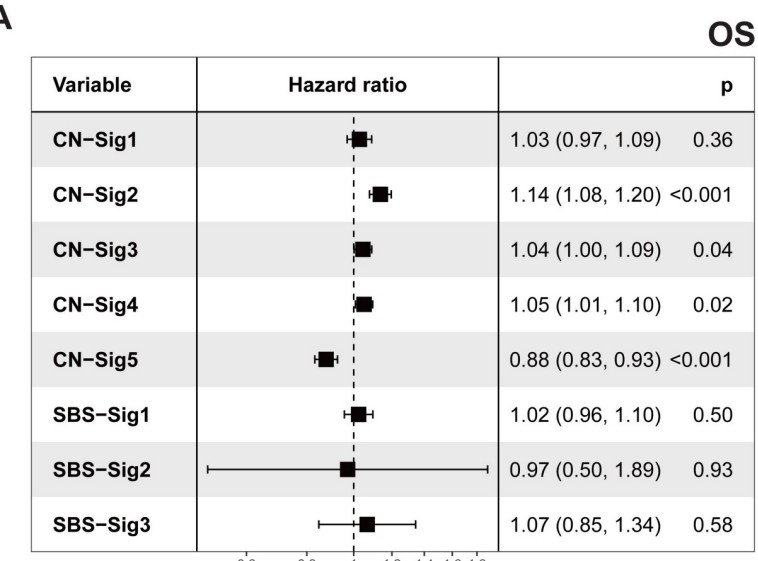

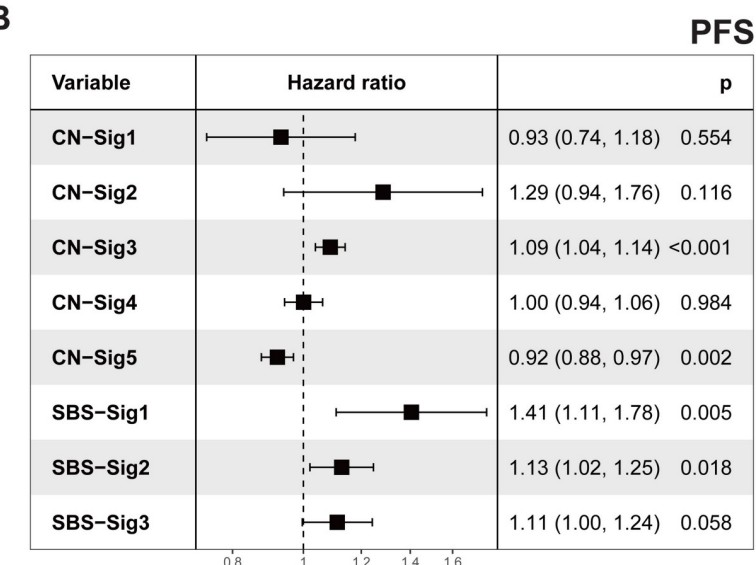

**Fig 6. Genome alteration signature exposures and PC survival.** (A,B) Forest plots showing the relative risk of signature exposures in overall survival (OS) (A) and progression-free survival (PFS). Due to the availability of clinical information, OS analyzes were performed with phs000178 dataset (TCGA, 498 primary prostate cancer) and phs000554 dataset (48 metastatic prostate cancer), PFS analyzes were performed with phs000178 dataset. (B). Exposures for all signatures were normalized to 1–20 for evaluating the hazard ratios per 5% exposure increase. Hazard ratios and p values were calculated by univariable Cox analysis. Squares represent hazard ratio, horizontal lines indicate the 95% confidence interval.

The associations between other clinical genomic parameters and OS, PFS are analyzed with univariate Cox regression method (S12 Fig). As expected, high Gleason scores are significantly associated with both poor OS and poor PFS. Late clinical stages are significantly associated with poor PFS. CNA burden is significantly associated with poor PFS.

This survival time analysis suggests that copy number signatures are associated with PC patients' survival. CN-sig 5, which has the lowest CNA burden, is associated with improved survival, while CN sig-2 is associated with poor survival. This is consistent with previous report that increased CNA burden was associated with poor PC survival [9]. In addition, this study further suggests that specific type of CNA represented by CN-sig 2 is associated with poor PC survival. CN-sig 2 has the highest hazard ratio among all five copy number signatures in both OS and PFS analysis. Of note, CN-sig 2 is not the copy number signature with the highest CNA burden (Fig 6).

## Discussion

Here a novel copy number alteration signature analysis method has been constructed. This method is featured by enhanced signature extraction efficiency, enhanced component visualization and comprehension, and can be coherently integrated with the known SBS signature analysis method. With this newly developed bioinformatics tool "sigminer", we performed the first copy number signature analysis in PC, and identified five copy number signatures. The mutational processes underlying each copy number signature have been illustrated. Copy number signatures show improved performance in PC patients' stratification and survival prediction compared with SBS signatures. Specific copy number signatures associated with PC metastasis and survival have been identified. These analyses provide novel insight into the etiology of PC, and novel biomarkers for PC stratification and prognosis. We demonstrated the performance of "sigminer" in PC, and this tool can be applied in other cancer types for the deep understanding of copy number alterations.

In survival analysis, the prognostic performance of CNA signature is much improved compared with SBS signature. In addition, copy number signature also shows improved association with PC metastasis and clinical stages compared with SBS signature. This implicates that the factors underlying copy number alterations can be major driving forces for PC progression and metastasis. And this difference needs to be further investigated in other cancer types. In addition, different types of copy number signature can have different impacts on PC prognosis, this greatly extends previous understanding that CNA burden is associated with PC prognosis [9,13].

Our copy number signature analysis tool "sigminer" only needs absolute copy number profile derived from WES or SNP array data, and high coverage WGS data is not needed. This is different from the recently described rearrangement signatures [29], which require high coverage WGS data for analysis. Copy number signature analysis can ultimately improve cancer patients' stratification, clinical prognosis and therapeutic outcome prediction.

## Methods

### Cohort collection and preprocessing

PC samples were included in this study when tumor and matched germline whole exome sequencing (WES) raw data (BAM or SRA files) from fresh frozen samples were accessible. The preprocessing steps are abstracted in S2 Fig. In total, 6 cohorts are included in this study, their dbGaP accession numbers are phs000178 (TCGA) [20], phs000447 [21], phs000554 [10], phs000909 [22], phs000915 [23] and phs001141 [24]. In total 1003 tumor-normal paired WES data are available for this study. BAM files for TCGA cohort were downloaded from GDC

portal (https://portal.gdc.cancer.gov/) with tool gdc-client. The average sequencing depth of each BAM file was summarized in S5 Table. SRA files for other cohorts were obtained from dbGaP database (https://dbgap.ncbi.nlm.nih.gov/) and converted to FASTQ files by SRA toolkit. Adapters were removed from the FASTQ files by trimgalore (https://github.com/FelixKrueger/TrimGalore). BWA MEM algorithm were then applied using hg38 as reference genome [30]. The result SAM files were converted to BAM files with samtools [31], followed by Picard toolkit (https://broadinstitute.github.io/picard/) to sort BAM files and mark duplications for variant and copy number calling. The corresponding codes are provided in section "Code availability".

### Clinical data

Clinicopathological annotations for cohorts phs000447, phs000554, phs000909, phs000915 and phs001141 were obtained from the original papers and dbGap database. Clinicopathological annotations for TCGA prostate cohort were downloaded from UCSC Xena by R package UCSCXenaTools v1.2.10 [32]. The data were cleaned and organized by in-house R scripts available in "Code availability" section.

### Absolute copy number calling from WES

Two bioinformatics tools including Sequenza [18] and FACETS [19] were performed to generate absolute copy number profiles from tumor-normal paired WES BAM files. For Sequenza, we followed its standard pipeline described in its vignette (https://cran.r-project.org/web/packages/sequenza/vignettes/sequenza.html) but with parameter 'female' set to 'FALSE' and a modified R package copynumber (https://github.com/ShixiangWang/copynumber) to work with hg38 genome build. For FACETS, we strictly followed its standard workflow described in its vignette (https://github.com/mskcc/facets/blob/master/vignettes/FACETS.pdf). Samples failed in calling were excluded from downstream analysis. In total Sequenza successfully called 937 pairs of WES data, and FACETS called 933 pairs of WES data. Results from FACETS and Sequenza are compared, and false positive signals have been carefully filtered. The somatic absolute copy number profiles detected in this study were summarized in S6 Table.

### Variant calling from WES

Somatic mutations including SBS and INDEL were detected by following the best practices of Genome Analysis Toolkit (GATK v4.1.3) with Mutect2 [33]. For comparison and data validation, additional variant caller, including varscan2 [34], muse [35], somaticsniper [36], have been applied. The lists of genome variants that passed all quality control filters were converted into VCF files by VCFtools [37]. The resulting VCF files were annotated by VEP [38] and further converted to MAF file by vcf2maf.pl (https://github.com/mskcc/vcf2maf). The MAF file was loaded into R, analyzed and visualized by Maftools [39]. The somatic genome variants detected in this study are available: https://zenodo.org/record/4591005.

### Genome alteration signature extraction procedures

To identify SBS and copy number signatures in a similar way, the analysis procedure has been properly abstracted into the following four common steps and implemented as an R/CRAN package sigminer (https://cran.r-project.org/web/packages/sigminer/).

1. Tally variation components. a) for SBS profile, same as previously reported [3, 4], for each tumor, we firstly classified mutation records into six substitution subtypes: C>A, C>G, C>T, T>A, T>C, and T>G (all substitutions are referred to by the pyrimidine of the

mutated Watson–Crick base pair). Further, each of the substitutions was examined by incorporating information on the bases immediately 5' and 3' to each mutated base generating 96 possible mutation types (6 types of substitution * 4 types of 5' base * 4 types of 3' base). b) For copy number profile, we firstly computed the genome-wide distributions of 8 fundamental copy number features for each tumor: the breakpoint count per 10 Mb (named BP10MB); the breakpoint count per chromosome arm (named BPArm); the copy number of the segments (named CN); the difference in copy number between adjacent segments (named CNCP); the lengths of oscillating copy number segment chains (named OsCN); the log10 based copy number segment size (named SS); the minimal number of chromosome with 50% copy number variation (named NC50); the burden of chromosome (named BoChr). These features were selected as hallmarks of previously reported genomic aberrations like chromothripsis or to denote the distribution pattern of copy number events [16,17,40,41]. The former 6 features have been used in previous study [15] to uncover the mutational processes in ovarian carcinoma. Next, unlike previous study[15], which applied mixture modeling to separate the 6 copy number features' distributions into mixtures of Poisson or Gaussian distributions, we directly classified 8 copy number features' distributions into 80 components according to the comprehensive consideration of value range, abundance and biological significance (S5E Fig). Most of the copy number components are discrete values, and the remaining are range values (see S1 Table). Based on the genome alteration component definition described above, two tumor-by-component matrices (one for SBS and the other for copy number) were generated and treated as the input of non-negative matrix factorization (NMF) algorithm for extracting signatures as previously reported, individually [3,4,42].

2. Estimate signature number. Signature number (or factorization rank) is a critical value which affects both the performance of NMF algorithm and biological interpretability. A common way for determining signature number is to try different values, compute some quality measures of the results, and choose the best value according to this quality criteria [42]. The most common criteria is the cophenetic correlation coefficient [43]. As suggested, performing 30–50 runs is considered sufficient to obtain a robust estimation of signature number value [42]. We performed 50 runs for both SBS (signature number range from 2 to 10) and copy number signatures (signature number range from 2 to 12). According to the cophenetic vs signature number plot (S5a and S5b Fig), the number of SBS signatures and copy number signatures were selected to be three and five respectively.

3. Extract signatures. After determining the signature number, we performed NMF with 50 runs to extract signatures for downstream analysis.

4. Quantify signature exposure (or signature activity). Sigminer package can provide both relative and absolute exposures of each signature for a tumor. The relative exposure of each signature represents its contribution proportion in a tumor among all signatures and can be directly obtained after NMF. The absolute exposure for each signature across tumors was determined after a scaling transformation as previously described [44]. For SBS signature, the absolute exposure represents the expected number of mutations associated with each SBS signature. For copy number signature, the absolute exposure represents the expected number of copy number segment records associated with each copy number signature.

## Signature profile normalization

Signature profile is essentially a matrix with row representing signature components and column representing the contributions of each component. Same as previous reported [4], SBS signature profile was normalized within each signature (i.e. by row), and a component value can be compared to another component value of the same copy number feature within each signature.

## Single sample copy number signature analysis and cancer subtype classification

After *de novo* signature discovery, relative and absolute signature exposures can be readily obtained with sigminer package. In clinical practice, it would be useful to quantify the signature exposures for a single tumor based on existing signatures (e.g. signatures from *de novo* signature discovery or public signature database). Sigminer package adopted the quadratic programming method to fit the existing signatures to one or more tumors. To classify a single tumor based on signature exposures, we built a 5-layer neural network model (input layer + hidden layer + 2 dropout layers + output layer) with Keras library (https://keras.rstudio.com/) and trained it with datasets used in our study. The evaluation and selection of hyperparameters of models were determined by grid search. This model can predict the subtype for single PC sample with copy number data available in high accuracy (average >0.9, see the link below). All steps to build and train the model for practical use accompanying with model training and validation results are packaged and available at https://github.com/ShixiangWang/sigminer.prediction for research community.

## Association analysis

Same as previously reported [15], associations between the exposures of signatures and other clinical or genomic features was performed using one of two procedures: 1) for a continuous association variable (including ordinal variable like clinical stage), Pearson correlation was performed; 2) for a binary variable, patients were divided into two groups and a Mann-Whiney U-test was performed to test for differences in average exposures of signatures between the two groups.

## Correlation network analysis

To investigate the structure of signature associations, correlation network analysis was performed with R package corrr (https://github.com/tidymodels/corrr) with continuous association variables. Variables that are more correlated appear closer together and are joined by thicker curves. Red and blue curves indicate positive and negative correlation respectively. The proximity of the points was determined with multidimensional clustering.

## Score definitions

To quantify the tandem duplication (TD) status for each tumor, we defined a tandem duplication phenotype (TDP) score, which capture the distribution and length information of TD across chromosomes [17,45]. This TDP score is calculated as:

$$\frac{TD_{total}}{\Sigma_{chr}|TD_{obs} - TD_{exp}| + 1} \times L$$

Where $TD_{total}$ is the total number of TD in a sample, $TD_{obs}$ and $TD_{exp}$ are the observed and expected TD for a chromosome in the tumor, $L$ is the total size of TD in Mb unit. TD is defined as copy number amplification segments with size range from 1Kb to 2Mb.

To quantify the chromothripsis state for each tumor, we defined a chromothripsis state score. This score is based on one key feature of chromothripsis, which forms tens to hundreds of locally clustered segmental losses being interspersed with regions displaying normal (disomic) copy-number [16]. This chromothripsis state score is calculated as:

$$\sum_{chr} N_{OsCN}^2$$

Where $N_{OsCN}^2$ is the square of total number of copy number fragments with absolute copy number value following "2-1-2" pattern. This score is a simplified estimation of chromothripsis state for each tumor and was visually inspected.

## Survival analysis

Univariate Cox analysis and visualization were performed by R package survival and ezcox (https://github.com/ShixiangWang/ezcox). The cox model returns hazard ratio value (including 95% confidence interval) per variable unit increase. To properly determine the relevance between variables and PC patients' OS (overall survival) and PFS (progression-free survival), exposures of all signatures were normalized to 1–20 for indicating the hazard ratio per 5% exposure increase. In a similar way, we multiplied some variables including CNA burden, Ti fraction (transition fraction) and tumor purity by 20 (the raw values of these variables range from 0 to 1). For other variables, we directly evaluated the hazard ratio per variable unit increase, e.g. hazard ratio for n_SBS indicates the hazard ratio per SBS count increase in a sample.

## Statistical analysis

Correlation analysis was performed using the Pearson method. Mann-Whiney U-test was performed to test for differences in signature exposure medians between the two groups. ANOVA was performed to test for differences across more than 2 groups. Fisher test was performed to test the association between signatures and categorical variables. In larger than 2 by 2 tables, Fisher p-values were calculated by Monte Carlo simulation. For multiple hypothesis testing, p values were adjusted using the false discovery rate method. All reported p-values are two-tailed, and for all analyses, $p < = 0.05$ is considered statistically significant, unless otherwise specified. Statistical analyses were performed by R v3.6 (https://cran.r-project.org/).

## Supporting information

**S1 Table. Copy number component parameter setting.**
(XLSX)

**S2 Table. Cohort characteristics.**
(XLSX)

**S3 Table. Summary statistics for somatic genome variations.**
(XLSX)

**S4 Table. Selected pathways and genes for association analysis.**
(XLSX)

**S5 Table. Sequencing depth information for BAM files.**
(XLSX)

**S6 Table. Summary table for absolute copy number profiles detected in this study.**
(XLSX)

**S1 Fig. Application comparison of two copy number signature extraction methods (our method and Macintyre et al. method) to ovary cancer with different sample size.** (A) The workflow of this analysis. Copy number data with different sample size are inputted into sigminer for matrix generation and signature extraction using two methods. (B, C) Signature profiles of different sample size generated through our method (B) and Macintyre et al. method (C) are shown. (D) Cosine similarity heatmaps for signatures extracted using our method. Cosine similarity analysis cannot be applied to signature profiles generated using Macintyre et al. method, because of unequal CNA components in ovarian cancer cohorts with different sample size.
(TIF)

**S2 Fig. Comparison between Macintyre et al. method and the new method developed in this study.** (A) Copy number signatures extracted with the Macintyre et al. method. (B) Representative copy number signature extracted with the new method in this study. The following differences are listed: (1) Here we used predefined copy number component, and calculate the weight of each component with a counting based method. Macintyre et al. applied mixture modeling to separate the copy number feature distributions into mixtures of Poisson or Gaussian distributions. The biological meaning of each component of Macintyre et al. method is not clear. While the biological meaning of our predefined component is clear. (2) Our method is much more computationally efficient than Macintyre et al. method. (3) Different normalization, we performed inter-feature row normalization, thus the values of the same signature can be compared, however, Macintyre et al. performed column normalization, and the values of the same signature cannot be compared. (4) Our method includes two additional copy number features "NC50" and "BoChr" to reflect the chromosome distribution pattern of copy number alterations. These features provided additional information for understanding copy number alterations. (C) List of the differences between Macintyre *et al* study and this study.
(TIF)

**S3 Fig. Study design and flowchart depicting the processing steps from raw sequencing data to genome alteration signatures.**
(TIF)

**S4 Fig. Mutational landscape of PC WES datasets.** Driver genes are identified by MutSig with q value < 0.05. The right panel indicates log10 based MutSig q values for driver genes. This plot was generated by Maftools with default setting.
(TIF)

**S5 Fig. Summary for small scale variants and copy number alteration features in PC WES dataset.** (A) Number of variants of different types in PC WES datasets. TNP: triple nucleotide polymorphism; SNP: single nucleotide polymorphism; INS: insertion; DNP: double nucleotide polymorphism; DEL: deletion. (B) Number of variants in each sample as a stacked barplot and variant classification as a boxplot. (C) Top 10 mutated genes as a stacked barplot by variant classification. (D) Length distribution of somatic copy number alterations (SCNA) segment and chromosome distribution of SCNAs. Location 'pq' represents a segment across both p arm and q arm. (E) Frequency distribution of 8 copy number features in combined PC WES

dataset. The x coordinate 23 in feature 'BoChr' represents chromosome X.
(TIF)

**S6 Fig. Cophenetic plot for determining the number of signatures.** (A,B) Cophenetic plot analysis for copy number signatures (A) and SBS signatures (B). (C,D) The consensus matrices for copy number signatures (C) and SBS signatures (D).
(TIF)

**S7 Fig. SBS signatures identified in PC and the workflow for signature stability analysis.** (A) Three SBS signatures are identified from PC WES data. Each signature was row normalized. (B) Bootstrap analysis workflow for signature stability applied to each tumor. For a tumor, firstly a mutation catalog was obtained based on component classification (96 components for SBS and 80 components for CNA), then a bootstrap procedure was adopted to get 1000 bootstrap catalogs based on observed mutation probability, this can generate a distribution of signature exposure for a p-value calculation in statistical test, and also quantify the signature stability by calculating RMSE between observed signature exposures (from observed catalogs) and bootstrap signature exposures (from bootstrap catalogs).
(TIF)

**S8 Fig. Cosine similarity analysis between SBS signatures identified in this study and COSMIC signatures.**
(TIF)

**S9 Fig. Correlation network analysis for genome alteration signatures.** Variables that are more highly correlated appear closer together and are joined by stronger curves. Red color indicates positive correlation and blue color indicate negative correlation. The proximity of the points are determined using multidimensional clustering. Associations with Pearson correlation coefficient r>0.2 are shown.
(TIF)

**S10 Fig. Sample clustering and heterogeneity analysis of PC based on the exposures of SBS signatures.** (A) For each PC patient, the relative contribution (bottom panel) and estimated SBS mutational counts (top panel) of each signature are shown as a staked barplot. PC samples are clustered into three groups based on the consensus matrix from multiple NMF runs, and each group is specified by one enriched SBS signature. (B) Quantification comparison for somatic and clinical parameters among each SBS signature enriched PC group by boxplot. The boxplot is bounded by the first and third quartile with a horizontal line at the median. ANOVA p values are shown. Abbr.: n_, number of, e.g. n_SNV, number of SNV; INDELs, small insertions and deletions; Ti_fraction, transition fraction; Tv_fraction, transversion fraction; Amp, copy number segments with amplification; Del, copy number segments with deletion; TDP score, tandem duplication phenotype score; cnaBurden, copy number alteration burden; MATH, MATH score is a quantitative measure of intra-tumor heterogeneity.
(TIF)

**S11 Fig. Associations between genome alteration signatures and PC clinical variables including metastatic status, clinical stage and Gleason score.** (A) Relationship between copy number signature exposure or SBS signature exposure and metastasis is shown by a Sankey plot. The sample size are indicated on the left of plot. (B,C) Fraction changes of clinical variables including metastasis, clinical stage and Gleason score in each copy number signature enriched PC groups (B) or each SBS signature enriched PC groups (C). p values were calculated by Fisher test with Monte Carlo simulation.
(TIF)

**S12 Fig. Genomic and clinical features and PC survival.** Forest plots showing the relative risk of selected genomic and clinical features in overall survival (OS) (A) and progression-free survival (PFS) (B). Due to the availability of clinical information, OS analyzes were performed with phs000178 dataset (TCGA, 498 primary prostate cancer) and phs000554 dataset (48 metastatic prostate cancer), PFS analyzes were performed with phs000178 dataset. Hazard ratios and p values were calculated by univariable Cox analysis. Squares represent hazard ratio, horizontal lines indicate the 95% confidence interval. Abbr.: n_, number of, e.g. n_SNV, number of SNV; INDELs, small insertions and deletions; Ti_fraction, transition fraction; Tv_fraction, transversion fraction; Amp, copy number segments with amplification; Del, copy number segments with deletion; TDP score, tandem duplication phenotype score; MATH, MATH score is a quantitative measure of intra-tumor heterogeneity.
(TIF)

## Acknowledgments

We thank Raymond Shuter for editing the text. Thank ShanghaiTech University High Performance Computing Public Service Platform for computing services.

## Author Contributions

**Conceptualization:** Xue-Song Liu.

**Data curation:** Shixiang Wang, Huimin Li, Minfang Song, Ziyu Tao.

**Formal analysis:** Shixiang Wang, Xue-Song Liu.

**Funding acquisition:** Xue-Song Liu.

**Investigation:** Shixiang Wang, Tao Wu, Zaoke He, Xiangyu Zhao, Xue-Song Liu.

**Methodology:** Shixiang Wang, Xue-Song Liu.

**Project administration:** Xue-Song Liu.

**Resources:** Kai Wu, Xue-Song Liu.

**Software:** Shixiang Wang.

**Supervision:** Xue-Song Liu.

**Visualization:** Shixiang Wang.

**Writing – original draft:** Xue-Song Liu.

**Writing – review & editing:** Xue-Song Liu.

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
