## [Decision Letter · Decision Letter 0]

22 Feb 2021

Dear Dr Liu,

Thank you very much for submitting your Research Article entitled 'Copy number signature analysis tool and its application in prostate cancer reveals distinct mutational processes and clinical outcomes' to PLOS Genetics.

We apologize for longer than usual time that took to find expert reviewers, which is likely due to current unusual pandemics  situation. The manuscript was now fully evaluated at the editorial level and by three independent peer reviewers. The reviewers appreciated the attention to an important problem, but raised some substantial concerns about the current manuscript. Altogether, we agree with reviewers that additional analyses and comparisons with results obtained with already published tools are required to further consider your work for PLOS Genetics.  Based on the reviews, we will not be able to accept this version of the manuscript, but we would be willing to review a much-revised version. We cannot, of course, promise publication at that time.

If you decide to revise the manuscript for further consideration at PLOS Genetics, please aim to resubmit within the next 60 days, unless it will take extra time to address the concerns of the reviewers, in which case we would appreciate an expected resubmission date by email to plosgenetics@plos.org.

[LINK]

We are sorry that we cannot be more positive about your manuscript at this stage. Please do not hesitate to contact us if you have any concerns or questions.

Yours sincerely,

Dmitry A. Gordenin, Ph.D.

Associate Editor

PLOS Genetics

David Kwiatkowski

Section Editor: Cancer Genetics

PLOS Genetics

Reviewer's Responses to Questions

**Comments to the Authors:**

Reviewer #1: The authors present sigminer, a tool able to identify signatures of copy number alterations. They use it to study copy number and single base substitution signatures in prostate cancer. They identify five copy number signatures (CNA signatures) and three single base substitution signatures (SBS signatures) in 1003 samples. They describe the association between CN-Sig 2 (the second copy number signature) and prostate cancer metastasis and a significant association with poor overall survival. CN-Sig 5 is associated with improved overall survival. In addition, they cannot find a clear association between most clinical parameters in prostate cancer and the three SBS signatures. They conclude that CNA signatures are more useful for patient stratification compared to SBS signatures in prostate cancer.

I think that sigminer can be useful to easily analyse other cancer types.

Comments:

1-The authors compare their method to a previously published method (Macintyre et al.).

What is the relation between the five CNA signatures in this study and the seven CNA signatures described in Macintyre et al. in ovarian carcinoma? Are some of them similar? Considering fig S1A, they seem quite different.

2-One of the novelty of sigminer is that it supports single sample analysis. However, this functionality is particularly interesting when we have a set of reference CNA signatures. Are the 5 CNA signatures identified in prostate cancer present also in other cancer types? Can these 5 CNA signatures considered reference signatures?

Reviewer #2: Manuscript by Wang et al. describe a new method/algorithm for copy number signature analysis in colorectal cancer. As the authors suggest there is definitely a need for such an algorithm. The manuscript as written needs major revision. The language throughout needs extensive review and revision for clarity. too much mix of passive and direct statements. too much repetition. There needs to be sanity checks on the initial calculation of copy numbers and mutations.

Why not proceed with matching copy numbers from both facets and Sequenza? Why not follow something similar to the TCGA protocol which involves multiple callers and careful filtering to remove false positives?. what was the overlap with the TCGA data?

Did you check the tcga mutation calls with your own calls? What was the overlap? Why did you only use mutect2? Strelka, varscan2 etc?

Page 21: "Next, unlike previous study[15], which applied mixture modeling to separate the 6 copy number features’ distributions into mixtures of Poisson or Gaussian distributions, we directly classified 8 copy number features’ distributions into 80 components according to the comprehensive consideration of value range, abundance and biological significance (Figure S4e).”

What is the rationale behind this? What is the biological significance of the features? Explain. There is comparison of the two methods in figs1 but this is not clearly explained.

pg 23: “

SBS signature profile was normalized within

546 each signature (i.e. by row). We performed row normalization within each copy

547 number feature, so a component value can be compared to another component

548 value of the same copy number feature within each signature. In summary, we

549 performed row normalization for SBS signature profile and row normalization

550 within each feature for copy number signature.”

Repeating the same thing. Simplify.

Why neural network? How does this signature assignment compare to nonlinear least squares or iterative fitting/complete enumeration?

Page 9

"Macintyre et al employed a mixture modeling based method for copy number

124 component extraction, and this method cannot be directly applied to other

125 cancer types”

Why not? I see no reason that these methods cannot be applied to other cancers than ovarian. This is not clear enough to reasoning to justify the methods used.

Page 9. 143-145. This is almost word to word repeat of the method. Again, there is no explanation as to what consists of biological significance.

Page 10. para starting with line 155. very unclear. it turns out you actually compared to previous work. Clearly state that. at the same time all the relevant details are buried in the S1 fig legend. Move those details to main text. in addition, explain what is the clear and fixed biological meaning of these 80 features. Why did you only compare your signature 1 with the previous method in Fig S1? I thought there were 3 signatures based on your method. Since both methods used similar features an in depth discussion of the similarities and differences is clearly warranted.

Page 10 last para. is another repeat statement. why? I don't need to read in every section you created a user friendly bioinformatic tool.

Reviewer #3: In this study, Wang et al. create a bioinformatic tool called sigminer to extract copy number signatures from tumor sequencing data. They apply this tool to a WES prostate cancer dataset containing both primary and metastatic samples and make observations about underlying mutational processes and association with clinical outcomes.

In general, copy number signatures may represent a valuable tool alongside more established mutational signatures as well as recently described structural variant signatures (Glodzik et al. Nat Genetics 2017, Nik-Zainal et al., Nature 2016), both in terms of gleaning insight into underlying mutational processes of cancer and as clinical biomarkers. However, the major limitations of this study are: 1) lack of novelty and demonstration of how this approach represents a conceptual advance over the study by Macintyre et al. Nat Genetics 2018; 2) lack of clear evidence that the CN signatures authors have derived will lead to meaningful biological insights related to prostate cancer.

Addressing the points below would substantially strengthen the manuscript.

A. Because Macintyre et al have already introduced the concept of copy number signatures, it is important for the authors to compare their method and the Macintyre et al. method head-to-head in the same datasets. The authors make some statements to suggest that their method has some advantages but this is not clearly demonstrated on actual data and it is unclear if these differences are substantive.

B. The authors should explain how their cohort of prostate cancer samples were chosen. Based on the references cited (#10, #20-24), this represents quite a heterogeneous set of samples both in terms of treatment stage (reference 10 represents CRPC, 24 PDXs that are enzalutamide resistant, 23 represents CRPC that is a mixture of enzalutamide-naïve and resistant, and 22 represents neuroendocrine prostate cancer, and 20-21 are localized prostate cancer). In particular, the use of samples from reference 22 is potentially problematic as neuroendocrine PCa has a very different biology and mutational spectrum than adenocarcinoma and should not be mixed for analysis. Would recommend that the authors use the ~1000 WES samples in the Armenia et al. Nature Genetics 2018 study for a more uniformly curated cohort. The use of this heterogeneous cohort may have significant influence on the clinical correlations obtained in the current version of the manuscript.

C. Most copy number changes are really a result of underlying structural variants (rearrangements). The authors need to discuss their method in the context of recently described rearrangement signatures (Nik-Zainal et al. Nature 2016, Menghi et al., PNAS, 2016, etc). One could argue that CN signatures and rearrangement signatures report on the same process. In that case, CN signatures could still be useful for certain data types (WES, cfDNA, shallow WGS) where RS derivation is not possible. But then the authors need to show this formally.

D. If the authors want to make claims about significance beyond this being a methods-only paper, signature analysis should be extended to other tumor types and/or validation datasets should be used for findings within a tumor type.

E. Points regarding specific signatures:

1) Can the authors comment on why SBS-2 and 3 are so similar?

2) The authors imply that CN-2 is a metastasis-specific biomarker because it is enriched in metastatic prostate cancer samples. But this is really a meaningful predictive finding if the authors can show that this signature is in primary prostate cancer that subsequently more likely to become metastatic.

3) If the authors plan to make a claim that CN-1 is associated with ecDNA this should be validated, if not experimentally, then at least with computational tools.

4) The fact that CDK12 mutations and other HRD mutations (BRCA1/2) are in the same signature is somewhat concerning. This is not consistent with rearrangement signatures in prostate and breast cancers where BRCA mutations are associated with small span size SVs than CDK12-associated duplications. Moreover, the clinical data from the TRITON study and others suggest that the response rate of CDK12-mutant prostate cancer to PARP inhibitors is virtually nonexistent, as compared with the response rate of BRCA mutant prostate cancer to these agents. So the current data do not suggest a role for CDK12 in homologous recombination repair in prostate cancer.

5) Overall, the use of CN signatures for meaningful risk stratification in prostate cancer and superiority over existing risk stratification schemes has not been clearly shown in this manuscript.

**Have all data underlying the figures and results presented in the manuscript been provided?**

Reviewer #1: Yes

Reviewer #2: Yes

Reviewer #3: Yes

PLOS authors have the option to publish the peer review history of their article (what does this mean?). If published, this will include your full peer review and any attached files.

Reviewer #1: No

Reviewer #2: No

Reviewer #3: No

---

## [Decision Letter · Decision Letter 1]

29 Mar 2021

Dear Dr Liu,

Thank you very much for submitting your Research Article entitled 'Copy number signature analysis tool and its application in prostate cancer reveals distinct mutational processes and clinical outcomes' to PLOS Genetics.

The manuscript was fully evaluated at the editorial level and by independent peer reviewers. The reviewers appreciated the attention to an important topic and thorough revisions done to the manuscript.  However,  reviewer 3 still have important concerns about significance of knowledge extension, data presentation and interpretation that we ask you to address in a revised manuscript.

We therefore ask you to modify the manuscript according to the review recommendations. Your revisions should address the specific points made by each reviewer by either making changes or explaining why changes and/or additional analyses are not needed.

[LINK]

Yours sincerely,

Dmitry A. Gordenin, Ph.D.

Associate Editor

PLOS Genetics

David Kwiatkowski

Section Editor: Cancer Genetics

PLOS Genetics

Reviewer's Responses to Questions

**Comments to the Authors:**

Reviewer #1: The authors addressed appropriately my two comments by presenting new analysis in S1 Fig.

Reviewer #2: The authors have addressed my questions.

Reviewer #3: In the revised manuscript, Wang et al. have made several textual clarifications and present new data in Figure S1 in which they perform CNA signature analysis in the TCGA ovarian cancer data using both their method and the Mcintyre et al. method.

Overall, while the authors have clarified some issues and sigminer seems like a reasonable new method for CN signature analysis, the manuscript remains largely similar to the first submission and my major reservations are primarily regarding the strength of biological insights that can be drawn at this stage, specifically:

1. Fig S1 – the authors run their method and the McIntyre et al. method on the TCGA ovarian cohort and report that there are some numerical differences (number of signatures extracted, number of components etc.). But what does this actually mean biologically? Can the authors comment on what underlying genomic processes the McIntyre signature is reporting on versus the authors’ signature and what are the reasons, if any, for the differences? Which is more faithful to what is known about the biology of the disease? The authors state repeatedly that cosine similarity cannot be performed between the two signatures but likely they can be compared in some other meaningful way(s).

2. Cohorts used for prostate cancer analysis – the authors provided clarification that the samples used are similar to those used in the Armenia study. This clarification is important but the point remains that the Armenia study included a mix of about 2/3 localized prostate cancer and 1/3 metastatic prostate cancers. If these are being treated together for Fig 6, S12 etc the conclusions may not be meaningful. Primary and metastatic prostate cancer are very different biologically and in terms of outcomes and should not be pooled into a single cohort.

3. Response to comment C – Understood that the CN analysis method can be run on WES and SNP array data while SV signatures require WGS data. My comment was intended to suggest that authors should/could run both CN and SV signatures on a WGS cohort so that they can comment on which CN/SV signatures are reporting on similar underlying mutational processes.

4. CN-Sig2 – the authors state that this signature contains both BRCA and CDK12 mutations as both are associated with tandem duplications. While this may be an explanation for why both mutations are enriched among CN-Sig2, this calls into question the biological significance of the authors’ CN signatures. We already know that BRCA and CDK12 mutations are different with respect to mechanism, the size of tandem duplications they cause, and the therapeutic vulnerabilities they unveil. Can the authors comment on the practical use of a CN signature that lumps these biologically two disparate mutations together?

5. Fig S12 – It seems to me like almost all the confidence intervals cross 1 except CNA burden which has a HR of 1.04 for OS and 1.05 for PFS, both almost crossing one. Can the authors confirm the following about this analysis?

a. How was CNA burden calculated and was it different than Hieronymous et al. elife 2018, which has shown that the HR for CNA burden in prostate cancer is on the order of ~1.4 in prostate cancer? Can the authors explain this discrepancy?

b. Was this HR in Fig S12 calculated on both primary and metastatic samples taken together? OS and PFS need to be assessed for each disease state individually.

c. Was this done as univariate or multivariable analysis?

**Have all data underlying the figures and results presented in the manuscript been provided?**

Reviewer #1: Yes

Reviewer #2: Yes

Reviewer #3: **No: **

PLOS authors have the option to publish the peer review history of their article (what does this mean?). If published, this will include your full peer review and any attached files.

Reviewer #1: No

Reviewer #2: No

Reviewer #3: No

---

## [Decision Letter · Decision Letter 2]

19 Apr 2021

Dear Dr Liu,

We are pleased to inform you that your manuscript entitled "Copy number signature analysis tool and its application in prostate cancer reveals distinct mutational processes and clinical outcomes" has been editorially accepted for publication in PLOS Genetics. Congratulations!

Yours sincerely,

Dmitry A. Gordenin, Ph.D.

Associate Editor

PLOS Genetics

David Kwiatkowski

Section Editor: Cancer Genetics

PLOS Genetics

Comments from the reviewers (if applicable):

Reviewer's Responses to Questions

**Comments to the Authors:**

Reviewer #2: The authors have addressed my questions.

**Have all data underlying the figures and results presented in the manuscript been provided?**

Reviewer #2: Yes

PLOS authors have the option to publish the peer review history of their article (what does this mean?). If published, this will include your full peer review and any attached files.

Reviewer #2: No

**Data Deposition**

http://datadryad.org/submit?journalID=pgenetics&manu=PGENETICS-D-20-01922R2

**Press Queries**

---

## [Editor Report · Acceptance letter]

29 Apr 2021

PGENETICS-D-20-01922R2 

Copy number signature analysis tool and its application in prostate cancer reveals distinct mutational processes and clinical outcomes 

Dear Dr Liu, 

We are pleased to inform you that your manuscript entitled "Copy number signature analysis tool and its application in prostate cancer reveals distinct mutational processes and clinical outcomes" has been formally accepted for publication in PLOS Genetics! Your manuscript is now with our production department and you will be notified of the publication date in due course.

With kind regards,

Andrea Szabo

PLOS Genetics

On behalf of:
